# Prevalence of dental caries and associated risk factors among People Living with HIV/AIDS and HIV uninfected adults at an HIV clinic in Kigali, Rwanda

Julienne Murererehe[1,2][☯]*, Yolanda Malele-Kolisa[2][☯], François Niragire[3][☯], Veerasamy Yengopal[4][☯]

1 Department of Preventive and Community Dentistry, University of Rwanda, Kigali, Rwanda, 2 Department of Community Dentistry, University of Witwatersrand, Johannesburg, South Africa, 3 Department of Applied statistics, University of Rwanda, Kigali, Rwanda, 4 Department of Community Dentistry, University of Western Cape, Cape town, South Africa

☯ These authors contributed equally to this work.
* jmurererehe@cartafrica.org

**Data Availability Statement:** All relevant data are within the manuscript and its Supporting Information files.

## Abstract

### Background

Dental caries is among the most frequent oral conditions in People Living with HIV/AIDS (PLWHA). There is a lack of baseline information on dental caries prevalence and associated risk factors among PLWHA in comparison to HIV uninfected people in Rwanda.

### Objective

This study was conducted to determine the prevalence of dental caries and associated risk factors among PLWHA and HIV uninfected adults at an HIV clinic of Kigali Teaching Hospital (CHUK) in Kigali, Rwanda.

### Methods

A comparative cross-sectional study was conducted among 200 PLWHA and 200 HIV uninfected adults aged 18 years and above attending an HIV clinic of CHUK. An oral examination was performed by a calibrated examiner. Dental caries were assessed using the WHO Decayed (D), Missing (M), and Filled Teeth (F) index (DMFT). Descriptive statistics, Chi-square, t-tests, and multiple binary logistic regression were used to analyze data.

### Results

Overall, a higher prevalence (50.5%) of PLWHA had experienced dental caries (DMFT>0) compared to HIV uninfected counterparts (40.5%) (p = 0.045). The prevalence of Decayed teeth (D) was also higher (23.5%) among PLWHA compared to HIV uninfected persons (13.6%) (p = 0.011). The Mean(SD) DMFT scores among PLWHA and HIV uninfected participants were 2.28 (3.68) and 1.29 (2.21) respectively (p = 0.001). After performing multiple binary logistic regression analysis, the predictors of dental caries in PLWHA were being

**Funding:** This study received funding from the Consortium for Advanced Research Training in Africa (CARTA) and the last is jointly led by the African Population and Health Research Center and the University of the Witwatersrand and funded by the Carnegie Corporation of New York (Grant No. G-19-57145), Sida (Grant No:54100113), Uppsala Monitoring Center, Norwegian Agency for Development Cooperation (Norad), and by the Wellcome Trust [reference no. 107768/Z/15/Z] and the UK Foreign, Commonwealth & Development Office, with support from the Developing Excellence in Leadership, Training and Science in Africa (DELTAS Africa) programme.

**Competing interests:** The authors have declared that no competing interests exist.

female (OR = 2.33; 95%CI = 1.14–4.75), frequent dental visits (OR = 4.50; 95% CI = 1.46–13.86) and detectable RNA viral load (OR = 4.50; 95% CI = 1.46–13.86). In HIV uninfected participants, the middle age range (36–45 years), and frequent dental visits were significantly associated with dental caries (OR = 6.61; 95%CI = 2.14–20.37) and (OR = 3.42; 95% CI: 1.337–8.760) respectively.

## Conclusion

The prevalence of dental caries was higher in PLWHA than in HIV uninfected counterparts. The reported higher prevalence of caries in PLWHA was associated with being female, detectable viral load, and frequent dental visits. Therefore, there is a need for effective oral health interventions specific to PLWHA in Rwanda to raise awareness of the risk of dental caries and provide preventive oral health services among this population. To ensure timely oral health care amongPLWHA, there is a need for an effort from policymakers and other stakeholders to integrate oral health care services within the HIV treatment program in Rwanda.

## Introduction

Oral health is an integral part of overall health and it is strongly associated with systemic conditions [1] including Human Immunodeficiency Virus (HIV) [2], Alzheimer's disease [3], cardiovascular diseases, diabetes and obesity [4]. The prevalence of caries is higher among People Living with HIV/AIDS (PLWHA) than among the general population [2,5–7]. Therefore, effective treatment of PLWHA should include oral diseases management.

Acquired Immune Deficiency Syndrome is considered an important public health problem in developed and developing countries [8]. More than 1.5 million people globally became newly infected with HIV in 2020 [9]. The WHO African region is the most affected region, accounting for more than two-thirds of PLWHAworldwide [8]. In Rwanda, HIV prevalence is higher in urban areas (7.1%) compared to rural settings (2.3%) [10].

Since the introduction of highly active antiretroviral treatment (HAART) in 1996, the life expectance of PLWHA has considerably increased [11,12]. HAART has also significantly reduced the occurrence of HIV related oral lesions [13]. As PLWHA live longer, it is imperative to promote their good oral health and access to quality dental care.

Dental care is reported as the least frequently met healthcare need among PLWHA [14]. Poor dental function resulting from oral diseases such as dental caries affect the general health of PLWHA [15,16]. Moreover, dental decay negatively affects the physical, mental, and social life of affected people [17]. In addition, oral diseases can affect adherence to antiretroviral therapy (ART) among PLWHA [17]. Also, dental problems among PLWHA are known to be more severe and difficult to manage compared to oral problems among HIV uninfected people [15]. Thus, oral health care should not be ignored and should form an important component of comprehensive treatment available for PLWHA.

There is a high burden of oral diseases in Rwanda according to the latest Rwandan National Oral Health Survey. Nearly 65% of participants in the survey had dental caries with more than 54% untreated caries [18]. Data on caries prevalence and associated risk factors among PLWHA is lacking in Rwanda. This study was conducted to provide baseline information on the prevalence of dental caries and associated risk factors among PLWHA in comparison to

HIV uninfected individuals at an HIV clinic of Kigali Teaching hospital (CHUK). The information from this study will enable us to inform policymakers and other stakeholders on the best strategies to prevent oral diseases, specifically dental caries among PLWHA thereby contributing to the improvement of their oral health-related quality of life.

## Materials and methods

### Study design and participants

This was a comparative cross-sectional study. The study population consisted of People Living with HIV/AIDS (PLWHA) and HIV uninfected adults attending the HIV clinic of Kigali Teaching Hospital (CHUK). All PLWHA, aged 18 years and older, and diagnosed with HIV infection at least 3 months prior to recruitment were included in the study. We also included all HIV uninfected attendees aged 18 years and above who came for HIV voluntary testing and who were diagnosed as HIV negative.

### Sample size and sample size calculation

The sample size for comparative cross-sectional study objectives was calculated using Stata software version 15. Assuming a study power of 80%, considering that the prevalence of caries is 60% in PLWHA and 45% in HIV uninfected persons and considering a ratio of 1 [19]. The minimum study sample estimated was 346 participants. Our study included PLWHA with the challenge regarding responsiveness especially in studies related to oral health that are not frequently done in Rwanda. For that reason, we decided to increase the calculated minimum sample by 20% to account for potential none respondents [20]. The study target became 415 participants of whom we successful collected data for 400 participants.

### Data collection tools and procedures

Participants were informed about the study and recruited at the two sites of the clinic as they attended for care on the recruitment day. All PLWHA and HIV uninfected participants at the HIV clinic of CHUK were outpatients. The first site of recruitment was at the Voluntary HIV Counselling and Testing (VCT) and it was for the recruitment of HIV uninfected adults. The second site was next to the physicians' rooms, and it was for recruiting eligible PLWHA at HIV clinic of CHUK. We worked hand in hand with physicians and practitioners who provided HIV results to the patients. To get HIV uninfected persons, respondents were first given their HIV status results by the nurse. After providing the results, the nurse informed people whose HIV results were negative about the ongoing research. HIV uninfected adults who showed interest to participate in the study, were sent to a data collection room to get informed consent. To recruit PLWHA, physicians/ nurses also first informed them about the ongoing study. PLWHA who showed interest to participate in the study were sent to the researchers'-room for consent signature before starting data collection.

Consecutive sampling method was used to select study participants. For data related to HIV infection (CD4 cell count, viral load, WHO staging, anti-retroviral treatment information), we used data extraction sheet to record the information needed from electronic medical record at HIV clinic. All consenting participants were examined for caries. The Clinical oral examination was performed by a calibrated experienced dental professional working at the University of Rwanda using the WHO (DMFT) index. Participants were seated in a semi-supine position and examined under natural light.

## Statistical analysis

Descriptive statistics including frequencies, percentages and means scores of oral diseases (DMFT mean cores) were computed. The Chi-square test was used to test the relationships between the presence of caries and categorical variables. A t-test was used for data associations between continuous variables. Binary logistic regression tests were done to determine the relationship of dental caries (dichotomous outcome) with various factors. A separate analysis was done with HIV related factors (CD4 counts, RNA viral load, type and duration on HIV treatment, WHO staging) as independent variables and adjusted for the sociodemographic, nutritional, and behavioral variables. Stata version 15 was used for the regression analysis. Kappa score statistical analysis was used to evaluate the intra-examiner reliability during examiner calibration and the kappa score value found was 0.74. The level of significance was set at 5%.

## Ethical consideration

The ethical clearances to conduct the study were received from Human Research Ethics Committee (No M200351) from the University of Witwatersrand, Institutional Review Board (No 573/CMHSIRB/2019) from University of Rwanda and Research ethics committee of Kigali Teaching Hospital (No EC/CHUK/026/2020). The informed written consents were given to all participants before data correction. To obtain informed consent, we first provided adequate information to participants through information sheet that explained in detail the nature and processes involved in the study, the reason for the study, and the intended outcomes. We gave participants opportunity to ask questions and we responded to each question. We ensured participants comprehended the provided information. Then, participants who agreed to participate in our study voluntarily signed the consent form. The confidentiality of patients was observed by using an anonymous questionnaire.

# Results

## Demographic characteristics

The demographic characteristics of People Living with HIV/AIDS (PLWHA) and HIV uninfected adults are presented in Table 1. The results revealed a statistically significant difference in age group among PLWHA compared to HIV uninfected individuals (p = <0.001). The mean age was significantly higher in PLWHA (43.51, 95%CI:41.51–45.51) than the mean age in HIV uninfected persons (36.53, (95% CI: 34.72–38.33) (p<0.001). Also, the majority of PLWHA were living in urban areas (81.5%) (n = 163) compared to HIV uninfected adults (63%) (p<0.001). More PLWHA completed primary (40%) and secondary school (34%) compared to HIV uninfected individuals who completed primary (37%) and secondary school level (25%) (p = 0.020). In addition, significantly (p = 0.024) more PLWHA were unemployed (32%) compared to unmployed HIV uninfected persons (22%). The prevalence of PLWHA in ubudehe (socio-economic status) category 3 and 4 was significantly higher (67.5%) compared to the prevalence of HIV uninfected respondents in ubudehe category 3 and 4 (51,5%) (p = 0.005).

## Comparison of dental caries among participants according to HIV infection status

The results in Table 2 show that PLWHA experienced significantly more caries than HIV uninfected individuals [(Decay (D) 47 (23.5%) vs. 27 (13.6%)]; p< 0.01]. The mean (SD) DMFT scores for PLWHA and HIV uninfected persons were 2.28 (3.68) and 1.29 (2.21) respectively, (p = 0.001).

**Table 1. Comparison of Socio demographic characteristics of participants.**

| Social demographic characteristics | PLWHA n(%); n = 200 | HIV uninfected n(%); n = 200 | P value |
|---|---|---|---|
| **Age** | | | **<0.001\*** |
| 18–24 | 30(15) | 39(19.5) | |
| 25–35 | 31(15.5) | 77(38.5) | |
| 36–45 | 34(17.0) | 39(19.5) | |
| 46–55 | 57(28.5) | 23(11.5) | |
| 56+ | 54(24.0) | 21(12.0) | |
| **Mean Age (mean, 95%CI)** | 43.51(41.51–45.51) | 36.53(34.72–38.33) | **<0.001\*** |
| **Sex** | | | |
| Male | 88(44.0) | 89(44.5) | 0.920 |
| Female | 112(56.0 | 111(55.5) | |
| **Residence** | | | |
| Urban | 163(81.5) | 126(63.0) | **<0.001\*** |
| Peri-urban | 25(12.5) | 32(16.0) | |
| Rural | 12(6.0) | 42(21.0) | |
| **Education** | | | |
| No formal schooling or less than primary school | 27(13.5) | 50(25.0) | **0.020\*** |
| Primary school completed | 80(40.0) | 74 (37.0) | |
| Secondary school | 68(34.0) | 50(25.0) | |
| Tertiary | 25(12.5) | 26(13.0) | |
| **Occupation** | | | |
| Unemployed | 64(32.0) | 44(22.0) | **0.024\*** |
| Employed | 136(68.0) | 156(78.0) | |
| **Use medical insurance** | | | |
| Yes | 199(99.5) | 195(97.5.0) | 0.10 |
| No | 5(2.5) | 22(11.0) | |
| **Ubudehe (SES)[+]** | | | |
| Category1 | 18(9.0) | 24(12.0) | **0.005** |
| Category2 | 47(23.5) | 73(36.5) | |
| Category3 and 4 | 135(67.5) | 103(51.5) | |

[+]Category 1: (Families who do not own a house and can hardly afford basic needs); Category 2: (Those who have a dwelling of their own or can rent one but rarely get one; Category 3: Those who have a job and farmers and go beyond subsistence farming to produce a surplus which can be sold. Also include those with small and medium enterprises who can employ dozens of people; Category 4: Those who own large-scale business, individuals working with international organizations and industries as well as public servants.

## Comparison of factors associated with dental caries among PLWHA and HIV uninfected adults

The results in Table 3 showed that females living with HIV/AIDS were 2.33 times more likely than males living with HIV/AIDS to have dental caries (95% CI = 1.142–4.252). Although not significant, HIV uninfected females were also 1.37 times more likely than HIV uninfected males to have dental caries 1.37 (95%CI = 0.680–2.774).

Amongst HIV uninfected adults, participants aged 36–45 years were 6.61 times more likely than the younger age group (18–14) to develop dental caries (95% CI = 2.146–20.370), the result was statistically significant (p = 0.001). The remaining age categories were not statistically significant.

HIV uninfected respondents who have visited a dentist within less than 6 months to 1-year period were 3.24 times more likely to have dental caries (95%CI = 1.34–8.76) compared to

**Table 2. Distribution of dental caries according to HIV infection status.**

| Variable | PLWHA (n = 200) | | HIV uninfected (n = 200) | | P-value |
|---|---|---|---|---|---|
| | n(%)Yes | n(%) No | Yes N(%) | n(%) No | 0.011* |
| D | 47(23.5) | 153(76.5) | **27(13.6)** | 172(86.4) | |
| M | 80(40) | 120(60) | 69(34.5) | 131(65.5) | 0.255 |
| F | 23(11.6) | 176(88.4) | 14(7) | 186(93) | 0.117 |
| DMFT | 101(50.5) | 99(49.5) | 81(40.5) | 119(59.5) | **0.045*** |
| | **Mean** | **SD** | **Mean** | **SD** | **P value** |
| D | 0.57 | 1.42 | 0.17 | 0.47 | **<0.001*** |
| M | 1.36 | 2.78 | **0.96** | 1.84 | 0.091 |
| F | 0.37 | 1.31 | **0.17** | 0.78 | 0.069 |
| DMFT | 2.28 | 3.68 | 1.29 | 2.21 | **0.001*** |

HIV uninfectedadults who never visited dentists. The difference was statistically significant. Participants who visited dentists within 5 years were approximately twice more likely to have dental caries (95%CI = 0.75–5.76) than the research participants who never received dental care. Those who visited dentist after 5 years were 1.30 times less likely to have dental caries (95% CI = 0.329–1.805) than those who never visited a dentist.

## Multiple binary logistic regression analysis of HIV related factors associated with dental caries among PLWHA

After adjusting for all factors not related to HIV infection (gender, age, residence, education, occupation, ubudehe (income status), dental visit, frequency of eating fruits, frequency of drinking tea with sugar and alcohol consumption), the results in Table 4 revealed that PLWHA with detectable RNA-Viral Load ($\geq$20) were 2.69 times more likely (95% CI = 1.004–7.208) to have dental caries than participants with undetectable RNA-Viral Load and the results were statically significant (p = 0.049).

## Discussion

The findings of this study revealed a significantly higher prevalence of dental caries in People Living with HIV/AIDS (PLWHA) compared to HIV uninfected individuals. Amongst PLWHA, cohort females, frequent dental visits, and detectable RNA-viral load were significantly associated with dental caries. In HIV uninfected participants, caries prevalence was associated with frequent dental visits and the age range of 36–45. The findings of this study are in line with growing evidence of a higher prevalence and risk of dental caries among PLWHA compared to the general population [5,21,22].

The higher prevalence of caries in PLWHA than in HIV uninfected individuals is similar to results by other scholars in developed and developing countries [2,5–7]. Untreated dental caries or decay (D) was also more prevalent in PLWHA 47(23.5%) than in HIV uninfected individuals 27(13.6%). This may be explained by the unavailability of oral health education and inaccessibility to dental care at HIV clinics. In addition, the lack of knowledge on oral health or low priority given to effect of oral conditions on the general health in Rwanda's HIV clinics can explain the higher prevalence of untreated dental caries among PLWHA. This is because, without knowledge, practitioners at HIV clinics cannot motivate or timely refer PLWHA to look for oral health services. Moreover, the shortage of oral health personnel, the lack of adequate dental infrastructures and unavailability of caries preventive programs for throughout

**Table 3. +Comparison of factors associated with dental caries among participants according to HIV status.**

| Variables | Subgroups | PLWHA OR (95% CI) | P value | HIV uninfected OR (95% CI) | P value |
|---|---|---|---|---|---|
| Gender | male | 1 | | 1 | |
| | Female | 2.33(1.142–4.752) | **0.020** | 1.37(0.680–2.774) | 0.375 |
| Age (years) | 18–24 | 1 | | 1 | |
| | 25–35 | 0.62(0.184–2.051) | 0.429 | 1.52(0.575–3.995) | 0.400 |
| | 36–45 | 0.50(0.160–1.577) | 0.239 | 6.61(2.146–20.370) | **0.001*** |
| | 46–55 | 0.74(0.261–2.068) | 0.561 | 3.48(0.953–12.689) | 0.059 |
| | 56+ | 2.04(0.650–6.412) | 0.221 | 2.66 (0.698–10.120) | 0.152 |
| Residence | Rural | 1 | | 1 | |
| | Urban | 0.95(0.239–3.787) | 0.944 | 1.39(0.577–3.341) | 0.463 |
| | Peri-urban | 1.18(0.230–6.060) | 0.840 | 1.03(0.324–3.287) | 0.955 |
| Education | | | | | |
| | Tertiary | 1 | | 1 | |
| | No or less than primary school | 0.93(0.244–3.557) | 0.919 | 1.92(0.575–6.415) | 0.288 |
| | Primary school completed | 0.92(0.329–2.581) | 0.877 | 0.93(0.294–2.951) | 0.905 |
| | Secondary school | 1.41(0.500–4.005) | 0.513 | 1.30(0.388–4.376) | 0.667 |
| Occupation | Not employed | 1 | | 1 | |
| | Employed | 0.98(0.478–2.007) | 0.957 | 2.12(0.860–5.220) | 0.102 |
| Ubudehe | Category1 | 1 | | 1 | |
| | Category 2 | 0.69(0.300–1.596) | 0.389 | 1.12(0.367–3.419) | 0.841 |
| | Category 3 and 4 | 0.92(0.285–2.994 | 0.896 | 1.07(0.357–3.189) | 0.907 |
| Dental visit | Never received dental care | 1 | | 1 | |
| | less than 6 months to 1 year | 4.50(1.460–13.865) | **0.009*** | 3.42(1.337–8.760) | **0.010*** |
| | more than 1 year but less than 5 years | 4.58(1.901–11.010) | **0.001*** | 2.08(0.754–5.761) | 0.157 |
| | 5 years and more | 1.53(0.694–3.358) | 0.292 | 0.77(0.329–1.805) | 0.550 |
| Frequency of eating fruits | several times a day or every day | 1 | | 1 | |
| | several times a week or once a week | 0.59(0.239–1.466) | 0.258 | 1.24(0.455–3.398) | 0.669 |
| | several times a month or seldom/never | 0.73(0.265–2.015) | 0.546 | 1.59(0.518–4.880) | 0.417 |
| Frequency of drinking tea with sugar | several times a month or seldom/never | 1 | | 1 | |
| | several times a day or every day | 2.02(0.891–4.590) | 0.092 | 1.85(0.770–4.344) | 0.169 |
| | several times a week or once a week | 2.37(0.889–6.290) | 0.084 | 1.01(0.360–2.834) | 0.984 |
| Alcohol consumption | No | 1 | | 1 | |
| | Yes | 1.59(0.748–3.393) | 0.227 | 1.63(0.785–3.373) | 0.190 |
| Fitting test result | | X | | XX | |

+The indicator of dental caries that we used in comparison with factors associated with dental caries was prevalence.

X Pseudo R2 = 0.131.

XX pseudo R2 = 0.141.

HIV clinics in the country can be associated with a higher prevalence of untreated dental caries among PLWHA.

Although Rwanda has shown remarkable improvement in HIV services and PLWHA can easily access ARVs, counseling services, regular checkups for non-communicable and other infectious diseases [23], no oral health services available at these specialized sevices. In addition, there are no regular prevention services, no program to raise awareness about dental caries and to inform PLWHA about dental caries risks and effect in existing HIV services or at the community level. This may contribute to a higher prevalence of dental caries among

Table 4. √Logistic regression analysis of HIV related factors associated with dental caries among PLWHA.

| Variable | Subgroups | OR (95% CI) | P value |
|---|---|---|---|
| CD4 | Group I (≥ 500) | 1 | |
| | Group II (200–499) | 0.904(0.440–1.856) | 0.784 |
| | Group III (<200) | 2.150(0.572–8.075) | 0.257 |
| RNA-Viral Load | Undetectable | 1 | |
| | Detectable | 2.691(1.004–7.208) | **0.049*** |
| WHO staging | Stage I | 1 | |
| | Stage II To IV | 2.130(0.658–6.895) | 0.207 |
| Types of ART | classII and III | 1 | |
| | Class I | 1.707(0.703–4.143) | 0.237 |
| Duration on ARVs | 11–15 years | 1 | |
| | 1–5 years | 1.089(0.313–3.788) | 0.893 |
| | 6–10 years | 0.988(0.436–2.237) | 0.978 |
| Duration of HIV-infection | 1–8 years | 1 | |
| | 9–16 years | 0.566(0.1552.059) | 0.388 |
| | 17+ years | 1.176(0.281–4.909) | 0.824 |
| Fitting test result | | • | |

√Adjusted for gender, age, residence, education, occupation, ubudehe (SES), dental visit, frequency of eating fruits, frequency of drinking tea with sugar and alcohol consumption.

• pseudo R2 = 0.174.

PLWHA than HIV uninfected persons. In their research, Feng and colleagues emphasized that oral health programs that target PLWHA should consider affordable dental care, a stigma-free setting, care delivered safely, and an accessible locations [24]. Thus, the need for an oral health program that can benefit PLWHA in Rwanda.

In contrast to the present study, in a research by Malele-Kolisa and colleagues, in South Africa, younger adolescents living with HIV had reported lower caries prevalence (D) than those of undiagnosed schools children [25]. This may be explained by contextual factors where there were oral health strategies and programs that targeted HIV infected adolescents and allowed them to have timely access to regular oral health education and preventive services in South Africa.

The Rwandan first National oral health strategic plan has highlighted barriers that may be associated with the burden of oral diseases in the country. The major part of oral health services is provided in District and referral hospitals. Very few primary health posts and centers can provide dental services. Many private dental clinics are mostly located in the City of Kigali and some other secondary cities. Primary oral health service is only provided in the form of analgesics for pain release in health posts and health centers and all patients are referred to hospitals for oral health care [26].

Since oral healthcare is provided in hospitals, it is not easy for the majority of Rwandans (who normally access primary health care at health posts and health centers level) to access oral health care due to the long distance to the hospitals. These issues result in the abandonment of oral health care for many patients. In addition, there is an overload of patients in hospitals with a shortage of oral health providers [26].

Moreover, in Rwanda District hospitals, there is a shortage of infrastructure including the lack of adequate space and functional dental equipment such as dental chairs to provide oral health services [26]. Apart from the recent and very first National strategic plan, there is no specific policy or plan that is specifically dedicated to oral health in Rwanda. Oral health is

integrated into different policy and regulatory documents related to None Communicable Diseases (NCDs) and there is no focal person in charge of oral health at the Ministry of health level to ensure the coordination and address issues related to oral health care in the country [26]. All those problems related to oral health services in Rwanda are not exceptional to PLWHA and since this group is more exposed to oral health problems including dental caries, it is very important to develop oral health interventions specific to PLWHA in Rwanda.

The results of our study also revealed a significant association between detectable HIV RNA-viral load and caries among PLWHA after adjusting for the other factors not related to HIV-infection. Similar to the results of our study, different researchers have also found an association between HIV RNA-viral load with dental caries [11,25,27,28]. These findings may suggest that dental caries can affect the immunity of PLWHA. However, based on the design of our study, it is not possible to determine the causal relationship between dental caries and higher RNA viral load among PLWHA. Thus, further studies are needed to give more clarifications on the mechanism of this relationship. On the other hand, PLWHA with immune suppression are reported to experience a reduction of saliva flow (xerostomia) [5]. The decreased saliva flaw contributes to easy development of dental caries due to lack saliva buffering capacity that controls the PH of mouth environment [29]. Based on the relationship of RNA viral load and dental caries, there is a need for oral health strategies to provide oral health education and early preventive dental services among PLWHA in Rwanda.

In our study, dental caries was significantly associated with being a female among PLWHA. These findings are similar to recent results of a study done in Uganda by Kalanzi and colleagues where HIV infected females had a higher prevalence of dental caries than HIV infected males [30]. Although not significant, HIV uninfected females were also more affected by dental caries than HIV uninfected males. The literature highlights different factors that expose women to dental caries than males in the general population. Women's family role has been shown as one of the factors for the higher prevalence of dental caries compared to males. Commonly, women have been family member with the responsibility for food preparation. This allows them to have access to foods and snacks between meals more frequently than males, which increases their chance of caries development compared to men [31,32]. For that reason, there is a need to develop vivacious oral health strategies that will target women in particular, especially in high-risk groups such as PLWHA.

Based on their exposure, women, especially those living with HIV/AIDS have to be given greater attention when planning preventive and oral health education interventions to decrease the risk of dental caries among this group. To increase the number of women who access dental services, previous interventions have considered home visits and oral health programs to provide oral education, and preventive oral health services [33]. Therefore, to increase accessibility to preventive oral health care among women living with HIV/AIDAS, there is a need for contextual oral health models among this population in Rwanda.

The results of our study revealed that a higher prevalence of PLWHA and HIV uninfected adults who frequently visited dentists experienced more caries compared to participants who less frequently visited dentists. Consistent results were reported in the literature [34,35]. As highlighted by other researchers, the DMFT index takes into account the past treatment history. Therefore, the DMFT index is affected by extracted teeth (M) and the fillings done due to dental caries (F). In our study, the M (Missing teeth due to caries) was the most prevalent component of DMFT in the group of PLWHA and HIV uninfected people. Having a higher prevalence of people with M component (extracted teeth) along with higher untreated caries among participants who frequently visited dentists suggests that participants in our study mostly received treatment for advanced dental caries leaving out caries that could be saved. This may suggest sub-optimal dental treatment in areas where our study participants visited dentists.

The sub-optimal dental care may be associated with a lack of conservative dental services in those areas or an overload of practitioners due to a shortage of dental personnel and lack of adequate infrastructure as highlighted in the recent first National oral health strategic plan in Rwanda [26]. Thus, there is a need to do further study to better understand factors associated with caries among participants who frequently visited dentists in Rwanda. To minimize the risk of progression of caries, there is a need to raise awareness and enforce caries risk assessment strategies for every dental patient in Rwanda especially among PLWHA.

Amongst HIV uninfected adults, middle age (36–45 years) was significantly associated with dental caries experience. A study from four countries (England and Wales, the United States, Japan, and Sweden) revealed a larger increase in DMFT in adulthood compared to the young group [36]. A recent study done in Ethiopia has also highlighted that age increase is significantly associated with caries experience [37]. However, some researchers had reported contrasting results where young age was associated with increased dental caries compared to adulthood [38]. These findings may be explained by various factors accross different community. For example, in many of developing countries, factors such as increase of sugary diet in young people, inavailability of preventive services, lack of awareness on how to do good oral hygiene and high cost of dental treatments expose young adults to caries than adults [37].

## Strength of the study

In Rwanda, this is the first study to look at the prevalence of dental caries and associated risk factors among PLWHA in comparison to HIV uninfected individuals. It is also among the few studies that compared PLWHA and HIV uninfected adults in regards to caries and its associated risk factors in Sub-saharan Africa.

## Limitations

The results of the study may not be generalizable to the general population of Rwanda because the study was done in an urban HIV clinic in Kigali city which could not represent the whole country. In addition, a cross-sectional study design makes it difficult to establish causality. Large longitudinal studies are needed to better understand the nature of the associations between caries and various risk factors. Our study was a comparative cross-sectional study that did not consider matching factors among two group of PLWHA and HIV uninfected participants. Although we adjusted for factors that were significantly different among the two group, we highly recommend further studies that will match key factors among the two groups to certainly compare PLWHA and HIV uninfected adults.

## Conclusion

After adjusting for key factors that were differenciating PLWHA and HIV uninfected participants, the prevalence of dental caries remained higher in PLWHA than in HIV uninfected participants. The reported higher prevalence of caries in PLWHA was associated with being female, detectable viral load, and frequent dental visits. Therefore, there is a need for effective oral health interventions specific to PLWHA in Rwanda to raise awareness of the risk of dental caries and provide preventive oral health services among Rwandan population especially PLWHA. To ensure timely oral health care among PLWHA, there is a need for effort from policymakers and stakeholders to integrate oral health care services within HIV treatment programs in Rwanda. Also, we recommend further studies that will consider the matching of key factors among PLWHA and HIV uninfected persons to ascertain the difference in dental caries among the two groups. Furthermore, there is a need to do further prospective studies that can

analyze in depth the strength of the association of caries with different factors among PLWHA in comparison to HIV uninfected individuals.

## Supporting information

**S1 Table. Description of participants' behaviors factors.**
(PDF)

**S2 Table. Description of other underlying conditions reported by participants.**
(PDF)

## Acknowledgments

We Thank Dr Babatunde Adedokun from the university of Ibadan for providing his expertise that greatly assisted data analysis, interpretation, and discussion of findings.

## Author Contributions

**Conceptualization:** Julienne Murererehe, Yolanda Malele-Kolisa, Veerasamy Yengopal.

**Data curation:** Julienne Murererehe, Yolanda Malele-Kolisa, Veerasamy Yengopal.

**Formal analysis:** Julienne Murererehe, Yolanda Malele-Kolisa, François Niragire, Veerasamy Yengopal.

**Funding acquisition:** Julienne Murererehe.

**Investigation:** Julienne Murererehe, Yolanda Malele-Kolisa, Veerasamy Yengopal.

**Methodology:** Julienne Murererehe, Yolanda Malele-Kolisa, François Niragire, Veerasamy Yengopal.

**Project administration:** Julienne Murererehe, Yolanda Malele-Kolisa, Veerasamy Yengopal.

**Resources:** Julienne Murererehe, Yolanda Malele-Kolisa, Veerasamy Yengopal.

**Software:** Julienne Murererehe, Yolanda Malele-Kolisa, François Niragire, Veerasamy Yengopal.

**Supervision:** Yolanda Malele-Kolisa, Veerasamy Yengopal.

**Validation:** Julienne Murererehe, Yolanda Malele-Kolisa, François Niragire, Veerasamy Yengopal.

**Visualization:** Julienne Murererehe, Yolanda Malele-Kolisa, François Niragire, Veerasamy Yengopal.

**Writing – original draft:** Julienne Murererehe, Yolanda Malele-Kolisa, François Niragire, Veerasamy Yengopal.

**Writing – review & editing:** Julienne Murererehe, Yolanda Malele-Kolisa, François Niragire, Veerasamy Yengopal.

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
