## [Decision Letter · Decision Letter 0]

2 Dec 2022

PONE-D-22-27201Prevalence of dental caries and associated risk factors among HIV-positive and HIV-negative adults at an HIV clinic in Kigali, RwandaPLOS ONE

Dear Dr. Julienne Murererehe,

Thank you for submitting your manuscript to PLOS ONE. After careful consideration, we feel that it has merit but does not fully meet PLOS ONE’s publication criteria as it currently stands. Therefore, we invite you to submit a revised version of the manuscript that addresses the points raised during the review process.

The reviewer(s) would like to see MAJOR revisions made to your manuscript before a final decision can be made. Therefore, I invite you to respond to the reviewer(s)' comments and revise your manuscript.

When you revise your manuscript please highlight the changes you make in the manuscript by using the track changes mode in MS Word or by using bold or coloured text.

We look forward to receiving your revised manuscript.

Kind regards,

Bingyu Liang

Academic Editor

PLOS ONE

Journal Requirements:

Reviewers' comments:

Reviewer's Responses to Questions

**Comments to the Author**

1. Is the manuscript technically sound, and do the data support the conclusions?

Reviewer #1: No

Reviewer #2: Yes

2. Has the statistical analysis been performed appropriately and rigorously? 

Reviewer #1: Yes

Reviewer #2: No

3. Have the authors made all data underlying the findings in their manuscript fully available?

Reviewer #1: Yes

Reviewer #2: Yes

4. Is the manuscript presented in an intelligible fashion and written in standard English?

Reviewer #1: No

Reviewer #2: Yes

5. Review Comments to the Author

Reviewer #1: The selection of the HIV negative control may have impacted of the results as the two populations were statistically different in a number of key factors.

Also, some conclusions - such as "the higher prevalence of caries experience among participants who

frequently visited dentists may suggest their poor oral healthcare-seeking behavior." They may be other factors making those with dental caries visit dentists more often.

Reviewer #2: The association between HIV and dental caries is an interesting subject. Regarding the article, I have the following questions that I would like to discuss with the author.

1. Please state in the manuscript whether the HIV negative participants had any other underlying diseases or caries related diseases. Otherwise, the sample is not representative when illustrating the association between HIV and dental caries

2. As mentioned in line 110 and 111, “those who agreed to participate were sent to the researchers for consent signature and data collection”, how to obtain the data of the participants other than dental examination, please describe in detail? Is it a self-answering questionnaire, a one-on-one interview, or something else?

3. Please state in the manuscript which software and version you used for your sample size calculations. According to the prevalence of caries in HIV+ patients and HIV- participants, the ratio and the study power provided by the manuscript, the sample size I calculated by PASS15.0 was a little different from yours.

4. Are there any missing values or no responses in your 400 samples, and if so, how do you consider them when calculating the sample size?

5. As mentioned in line 120 and 121, “Multiple logistic regression tests were done to determine the relationship of dental caries (dichotomous outcome) with various factors”, I don’t really understand why Multiple Logistics Regression was used instead of Binary Logistics Regression, as you yourself said that dental caries was dichotomous outcome. Could you please identify the dependent variables in the regression analysis?

6. What software and version were used for the regression analysis? They need to be stated in the manuscript.

7. As mentioned in line 124 and 125, “Kappa score statistical analysis was used to evaluate the intra-examiner reliability during examiner calibration”, so what was the value of the Kappa score? Please state it in the manuscript.

8. Why the category 3 and the category 4 were described together in Table 1?

From your description of categories 3 and 4, I think their SES are very different and should not be combined. Actually, the prevalence or DMFT of dental caries is strongly correlated with SES. If the objective of the study is to discuss the relationship between HIV and dental caries, the more balanced the distribution of samples in SES, the more bias can be reduced.

9. Please describe what analysis was performed in Table 3. What indicator of dental caries was used in comparison with factors associated with dental caries? Is it prevalence? It needs to be described in the manuscript.

10. The meaning of the sentence in this line of 178 and 179 “Participants who visited dentists after 1 year but less than 5 years” is unclear.

11. I don't quite understand the analysis method used by the authors in Table 3. If it was a single factor Chi-square analysis, why were HIV-related factors such as viral load didn’t need to be included in this analysis, but only in the regression analysis? If it was the result from a logistic regression, the total associated factors also should be included and the results should be presented in one table including both the Crude Odds Ratio and the Adjusted Odds Ratio.

12. The fitting test result of the logistic regression model should be attached at the end of Table 4, so that readers can judge the reliability of the model.

6. PLOS authors have the option to publish the peer review history of their article (what does this mean?). If published, this will include your full peer review and any attached files.

Reviewer #1: **Yes: **Harriet Mayanja-Kizza

Reviewer #2: No

---

## [Author Response · Author response to Decision Letter 0]

17 Jan 2023

January 15th, 2023

Dear Editor,

Re: Resubmission of Revised Version of Manuscript PONE-D-22-27201 for publication 

Please find attached a revised version of our manuscript PONE-D-22-27201

We would like to thank the editor and reviewers for the insightful comments that have greatly helped us to improve the quality of our manuscript. We have provided detailed responses to every comment from the academic editor and reviewers. For our responses, we used italic font and highlighted them in green color.

We hope that these revisions are sufficient to make our manuscript suitable for publication in Plos one and look forward to hearing from you at your earliest convenience

Sincerely,

Julienne Murererehe

KG 11 Ave Remara, School of Dentistry, University of Rwanda, Kigali, Rwanda.

Tel; +250788593017

Email: jmurererehe@cartafrica.org

Editor’s comments: 

Response 1: We observed the requirements for Plos one style in the revised manuscript. 

Response 2: We agreed with the editor’s comment and we thank him for pointing it out. The following is the funding information statement with the correct grant numbers for the awards received for the study: 

This study received funding from the Consortium for Advanced Research Training in Africa (CARTA) and the last is jointly led by the African Population and Health Research Center and the University of the Witwatersrand and funded by the Carnegie Corporation of New York (Grant No. G-19-57145), Sida (Grant No:54100113), Uppsala Monitoring Center, Norwegian Agency for Development Cooperation (Norad), and by the Wellcome Trust [reference no. 107768/Z/15/Z] and the UK Foreign, Commonwealth & Development Office, with support from the Developing Excellence in Leadership, Training and Science in Africa (DELTAS Africa) programme. 

The correct grant numbers for the awards received for your study are included in the cover letter.

Response 3: We changed the information regarding repository information for this research as follows: “All relevant data are within the manuscript and its Supporting information files”. (Refer to cover letter). 

Response to Reviewers’ Comments: 

Reviewer#1

Abstract

1. Line 17-19; “HIV-positive (HIV+) persons.” – Avoid hyphen between HIV positive and HIV negative phrase’s. Can be confusing as HIV-positive looks like a negative. 

Response 1: We thank the reviewer for this comment. We removed hyphens between HIV positive and HIV negative phrases on lines 17-19 and throughout the manuscript. 

2. Line 22 – attending an HIV clinic

Response 2: We thank the reviewer for the comment. The change was done as “attending an HIV clinic (line 27).

3. Line 27 – Dental Caries were assessed using the WHO Decayed (D), ……

 Response 3: We thank the reviewer for this comment. We did the amendment as follows:

 “Dental Caries were assessed using the WHO Decayed (D) (line 28)”.

4. Line 31 – delete experience

Response 4: We appreciate the reviewer’s comment. We deleted the word “experience”. We edited as follows: “HIV+ adults had experienced dental caries (DMFT>0)” (Line 32)

5. Edit to “being female”

Response 5: We Thank the reviewer’s observation. We addressed this comment as follows 

“being female” (line 37).

Introduction 

1. Line 51-52 – five an example of these “systemic conditions”

Response 1: We thank the reviewer for this comment. We added examples of systemic conditions as follow: “including Human Immunodeficiency Virus (HIV) (Birungi et al., 2022), Alzheimer’s disease (Tsuneishi et. al , 2021), Cardiovascular diseases, diabetes and obesity (Sabharwal, Stellrecht, & Scannapieco, 2021) (line 53-55).

2. Line 52 – citation does not describe Caries prevalence in HIV positive – it’s about stigma

Response 2: We thank the reviewer for this observation. We replaced the reference by more relevant ones (line 56).

3. Line 54 unclear

Response 3: We thank the reviewer for pointing out the lack of clarity. We rephrased the statement as bellow in the revised manuscript: “Therefore, effective treatment of HIV patients should include oral diseases management” (line 56-57).

Materials and Methods

1. Line 88 - HIV voluntary testing and who was “were, Not was” 

Response 1: We thank the reviewer for the careful observation. We edited the statement to

“HIV voluntary testing and who were (line 93)”

2. Line 100 onwards. I assume even HIV negative were outpatients. Please clarify

Response 2: We appreciate the reviewer’s comment. HIV negative were also outpatients and we addressed this as follow: “All HIV+ and HIV- participants at the HIV clinic of CHUK were outpatients” (line 110-111) 

3. Among the HIV positive participants, was informed consent obtained, and if so when was this done? Also clarify in lines 131 and 132 regarding process of obtaining informed consent

Response 3: We thank the reviewer for pointing out the incomplete information. Yes, informed consent was obtained among HIV+ participants. It was obtained prior to start data collection in the room allocated to data collectors. We addressed the comment as follow:

“HIV+ persons who showed interest to participate in the study were sent to the researchers ‘room for consent signature prior to data collection (line 116-117)”.

Regarding the process of obtaining informed consent, we clarified the process as follow: 

 To obtain informed consent, we first provided adequate information to participants through information sheet that explained in detail the nature and processes involved in the study, the reason for the study, and the intended outcomes. We gave participants opportunity to ask questions and we responded to each question. We ensured participants comprehended the provided information. Then, participants who agreed to participate in our study voluntarily signed the consent form (line 142-147). 

4. Line 104 – revise term “old cases” 

Response 4: We appreciate the reviewer’s insight and we revised the term as follow: 

“Eligible HIV+ persons” (line122-124)

5. How were participant selected? Was it consecutive, or some other criteria was used

 Response 5: We thank the reviewer for this comment. We responded to it as follow: “Consecutive sampling method was used to select participants” (line 121-122).

6. Was brushing teeth also considered as a factor. 

Response 6: We appreciate the reviewer insightful question. Brushing teeth were considered as factor (supporting information: S1 table). However, it was excluded from the regression analysis model because almost all participants (more than 91%) reported that they brush their teeth daily and that outcome could not provide relevant information on the outcome variability in regression analysis.

7. What of soft drinks and sweets among the foods other than just sugar and fruits. 

Response7: We Thank the reviewer the comment. Based on the WHO survey questionnaire, we also asked our participants information on taking lemonade, Coca Cola or other artificial juices (Supporting information: S1 table). The outcome of this information was not considered during our regression analysis based on the outcome variability in regression analysis. 

8. Table 4 – clarify – WHO staging for HIV

Response 8: We thank the reviewer for point out the lack of clarity regarding WHO staging for HIV+ participants:

WHO staging for HIV consisted of recording information on baseline assessment of HIV patients that informed the provision of ongoing care. We did not assess this information ourselves. However, we used data extraction sheet to record information on WHO staging for HIV+ participants from existing electronic medical record.

Based on the literature, WHO staging for HIV consists of the following stages: 

Stage one (primary HIV infection characterized by flu like illness (fever, diarrhea, weight loss) that lasts for few weeks). Stage two characterized by clinically asymptomatic stage that last several years (up to 10 years). Stage three consists of symptomatic HIV infection, damage to immune system and appearance of opportunistic infections. Stage 4 is the progression of HIV to AIDS. T cell count drops to 200 and opportunistic infections became worse

Results

1. Line 147 – what is “ubudehe” ??? – Only defined in line 191 for first time - ubudehe (income status),…

Response1: We agree with this observation and we thank the reviewer for pointing it out. We did the amendment by defining the term “ubudehe” right from beginning

Ubudehe (Social economic status) (line 165)

2. In table one – how would or did these differences in age, sex, education status etc. impact on prevalence of dental caries. 

Response 2: We appreciate the reviewer’s comment. In this study, HIV positive and HIV negative participants differ in regards to age (HIV+ were older than HIV-), education level (HIV+ were more educated than HIV-), employment (HIV- persons were more employed than HIV+) and income status (HIV+ presented higher income done HIV+). During our regression analysis we accounted for the difference in those factors by controlling the effect of age, education, employment and income status. HIV positive participants remained with higher likelihood of having dental caries than HIV negative individuals even after controlling for the above mentioned factors. 

3. For example, are they generally lower in younger more educated persons? What does the literature say regarding these factors in table one in a general HIV neg for example population? 

Were HIV positive living in urban middle class or urban low income earner settings (such as more of slum areas? 

Response 3: We thank the reviewer for the insightful comment. According to literature, older age, low education, being unemployed and low income status are associated with high prevalence of dental caries. To resolve the issues of those differences we removed the effect of these key factors during the regression analysis 

Regarding whether HIV positive were living in urban middle class or urban low income earner settings, there is not specific urban middle class or urban low income earner settings according to Rwanda context. Therefore, HIV positive participants were living either in urban, peri urban or rural area. 

4. Line 190-192 unclear context. Please clarify on this statement regarding {frequency of eating fruits, frequency of drinking 192 tea with sugar and alcohol consumption} in the multiple regression analysis. 

Response 4: We appreciate the reviewer’s comment. Among factors that we looked at in this study, we also asked questions related to diet habit including frequency of eating fruits, frequency of taking tea with sugar and frequency of alcohol consumption. By asking about frequency of eating fruits, we wanted to identify whether our participants consume healthy diet and how frequently (and this is a protective factor). We asked about the frequency of alcohol consumption because frequently consuming alcohol lead to dehydration with the risk of developing caries. Frequently consuming sugary tea is also a risk factor for dental caries. 

So when we were analyzing HIV-related factors, we accounted for all those previous factors not related to HIV-infection to remove their effect 

Discussion

1. The statistically significant differences between the HIV neg and HIV pos. populations seems to indicate that the two groups were not comparable. This makes if difficult to make meaningful conclusions of these results comparing the 2 patient populations. Could the controls be patients attending the dental clinic – who tested HIV negative rather than those attending the HIV clinic for an HIV test – who may be younger etc. 

Response 1: We thank the reviewer for the important comment. Based on the approach used in this study, it was not possible to know HIV status of our study participants from dental clinic. This is because HIV tests are not performed in dental clinics in Rwanda. Also it was not possible to move participants from Dental clinic to get HIV test at HIV clinic. Therefore, we recruited HIV negative participants from HIV clinics. However, during the analysis, we controlled the effect of all those key factors (age, education, employment, residence) that could have made it difficult to compare the two group (table3).

2. Data also showed that dental caries are more in older persons, and the HIV patients were significantly older. 

Response 2: We thank the reviewer for this comment. During regression analysis, we controlled the effect of age, still caries remained higher among HIV+ individuals compare to HIV negative counterparties. 

3. Lines 215 to 216 – It’s important to consider that factors other than dental hygiene education in HIV clinics – to explain the higher dental caries among the HIV infected persons. 

Response 3: We appreciate the reviewer’s comment that allowed us to improve our discussion. Untreated dental caries or decay (represented by D in DMFT) indicate existing cavities that were not treated. So we considered the reviewer’s comment and edited our discussion as follow: 

“This may be explained by the unavailability of oral health education and inaccessibility to dental care at HIV clinics. In addition, the lack of knowledge on oral health or low priority given to effect of oral conditions on the general health in Rwanda HIV clinics can explain the higher prevalence of untreated dental caries among HIV+ patients. This is because, without knowledge, practitioners at HIV clinics cannot motivate or timely refer HIV+ persons to look for oral health services. Moreover, the shortage of oral health personnel, the lack of adequate dental infrastructures and unavailability of caries preventive programs for HIV+ persons throughout HIV clinics in the country can be associated with a higher prevalence of untreated dental caries among HIV+ individuals” (line 238-246).

4. Could caries be more in women because of their health seeking behaviors? 

Response 4: We appreciate the reviewer for this kind and insightful comment. Based on the literature, females act more positively compared to males when it comes to oral health seeking behaviors (Su et. al , 2022). Therefore, based on our findings, we discussed other possible factors related to females’ risk to dental caries rather than health seeking behaviors. Also we remove the argument of poor oral health seeking behavior in our discussion (line 340-344)

5. Line 264 – women’s reproductive history was not determined, so difficult to attribute the higher dental caries to hormonal and reproductive factors. 

Response 5: We understand and agree with the reviewer’s comment and we thank the reviewer for this important comment. We amended the reviewer’s suggestion by removing the information related to hormonal and reproductive factors (307-309). 

6. Lines 276-277 –” The results of our study revealed that a higher prevalence of HIV positive and HIV negative adults who frequently visited dentists experienced more caries compared to participants who less frequently visited dentists.” It possible that those with cries may be more prone to dental caries – e.g. genetic or eating habits – so visit dentists more, rather than risk being higher – by visiting dentists.

Response 6: We appreciate the reviewer’s comments. We considered risk factors other than visiting dentistry itself. We amended as following: 

 “Having a higher prevalence of people with M component (extracted teeth) along with higher untreated caries among participants who frequently visited dentists suggests that participants in our study mostly received treatment for advanced dental caries leaving out caries that could be saved. This may suggest sub-optimal dental treatment in areas where our study participants visited dentists. The sub-optimal dental care may be associated with a lack of conservative dental services in those areas or an overload of practitioners due to a shortage of dental personnel and lack of adequate infrastructure as highlighted in the recent first National oral health strategic plan in Rwanda (MinistryofHealth, 2019). Thus, there is a need to do further study to better understand these factors associated with caries among participants who frequently visited dentists in Rwanda. To minimize the risk of progression of caries, there is a need to raise awareness and enforce caries risk assessment strategies for every dental patient in Rwanda.” (Line 325-340)

7. Also poor health seeking behavior is not compatible with the data of more dental visits. Is it possible that the dental care given is sub optimal – this possibly more visits for the same (rather than recurring) dental caries?

Response 7: We thank the reviewer and agree with the concern raised. We edited our discussion idea by removing poor oral health seeking behaviors as a risk factor for dental caries among participants who frequently visited dentistry (Line 341-345)

8. The viral load issue seems to be the single most significant outcome of this study – yet its discussed last

Response 8: We appreciated the reviewer’s comment. We amended it by moving the discussion on viral load to earlier paragraphs (line 284-296).

Limitations and conclusions

1. It important to mention the limitation of the non-comparable HIV pos. and neg demographic characteristics.

Response 1: We thank the reviewer for this comment. We added the following limitation regarding HIV positive and HIV negative comparison: 

“Our study was a comparative cross-sectional study that did not consider matching factors among two group of HIV positive and HIV negative participants. Although we adjusted for factors that were significantly different among the two group, we highly further studies that will matching key factors among the two groups to certainly compare HIV+ and HIV- adults” (line 379-383).

2. Also the above differences may impact on the study conclusions 

Response 2: We thank the reviewer for the comment. In our conclusion, we acknowledge the limitation of our study by recommending further studies that will consider the matching of key factors among HIV positive and HIV negative to ascertain the difference in dental caries among the two groups (line 396-397).

References

1. Ref 2 – use proper citation

Response 1: We appreciate the reviewer’ s comment and the citation 2 was replaced by the more relevant reference as suggested by the reviewer (line 54-55) 

2. Infect, all reference citation needs to be improved to one format. 

Response 2: We thank the reviewer. We updated all reference citations in Vancouver format as recommended in the journal guideline. 

Reviewer 2

The association between HIV and dental caries is an interesting subject. Regarding the article, I have the following questions that I would like to discuss with the author.

1. Please state in the manuscript whether the HIV negative participants had any other underlying diseases or caries related diseases. Otherwise, the sample is not representative when illustrating the association between HIV and dental caries

Response1: We thank reviewer for the insightful comment. We recorded information on any underlying diseases that HIV negative participants were suffering from to account for it during the regression analysis. A very small number of HIV negative reported having diabetes 7(3.50%), Hypertension 13 (6.50%) and only 1(0.50%) of them were pregnant. We did not consider the underlying diseases in regression model because that outcome could not provide relevant information on the outcome variability in regression analysis (Supporting information: S2 table).

 Variables HIV-

 n=200

 Yes n(%) No n(%) Not applicable* n(%)

Diabetes 7(3.50) 193 (96.50) -

Hypertension 13 (6.50) 187 (93.50) -

Being pregnant 1 (0.50) 109 (54.50) 90 (45.00)

* Not applicable indicates males who cannot be pregnant 

2. As mentioned in line 110 and 111, “those who agreed to participate were sent to the researchers for consent signature and data collection”, how to obtain the data of the participants other than dental examination, please describe in detail? Is it a self-answering questionnaire, a one-on-one interview, or something else?

Response 2: We thank the reviewer for this comment to clarify on how we obtained the data. 

We used one to one interviews to obtain data on risk factors not related to HIV-infection. For data related to HIV infection (CD4 cell count, viral load, WHO staging, anti-retroviral treatment information), we used data extraction sheet to record the information needed from electronic medical record at HIV clinic (Line 122-124).

3. Please state in the manuscript which software and version you used for your sample size calculations. According to the prevalence of caries in HIV+ patients and HIV- participants, the ratio and the study power provided by the manuscript, the sample size I calculated by PASS15.0 was a little different from yours.

Response 3: We appreciate the reviewer’s comment. We used Stata version 15 to calculate the sample size (line 136). 

4. Are there any missing values or no responses in your 400 samples, and if so, how do you consider them when calculating the sample size? 

Response 4: We Thank the reviewer for this comment. Our study included HIV positive participants with the challenging of low responsiveness rate especially in studies related to oral health that are not frequently done in Rwanda. For that reason, we took the decision to increase the calculated minimum sample by 20% to account for potential none respondents (Ntampaka , Nyaga, Niragire, Gathumbi, & Tukei, 2019). The study targeted 415 participants of whom we successful collected data for 400 (line 102-106). 

5. As mentioned in line 120 and 121, “Multiple logistic regression tests were done to determine the relationship of dental caries (dichotomous outcome) with various factors”, I don’t really understand why Multiple Logistics Regression was used instead of Binary Logistics Regression, as you yourself said that dental caries was dichotomous outcome. Could you please identify the dependent variables in the regression analysis?

Response 5: We thank the reviewer for point out this comment. We understand and agreed with the reviewer. We edited as follow: “The binary logistic regression tests were done to determine the relationship of dental caries (dichotomous outcome) with various factors” (line 132-133)

6. What software and version were used for the regression analysis? They need to be stated in the manuscript.

Response 6: We agree with the reviewer’s suggestion. We amended the suggestion as follow: “Stata version 15 was used for the regression analysis” (line 136) 

7. As mentioned in line 124 and 125, “Kappa score statistical analysis was used to evaluate the intra-examiner reliability during examiner calibration”, so what was the value of the Kappa score? Please state it in the manuscript.

Response 7: We appreciate the reviewer’s useful comment to clarify more the method process. The value of the Kappa was 0.74. We edited the manuscript as follow: “The kappa score value found was 0.74” (Line 138).

8. Why the category 3 and the category 4 were described together in Table 1? 

From your description of categories 3 and 4, I think their SES are very different and should not be combined. Actually, the prevalence or DMFT of dental caries is strongly correlated with SES. If the objective of the study is to discuss the relationship between HIV and dental caries, the more balanced the distribution of samples in SES, the more bias can be reduced.

Response 8: We Thank the reviewer for this important comment. During regression analysis, separating categories was not working and it was leading to empty results. Therefore, we combined SES categories based on the existing roles that they play on health insurance and other socio economic needs in Rwanda. In fact, participants in categories 1 and 2 are considered poor (line 171-172) and they are supported by the government to get health insurance and other social economic needs such as education. People in category 3 and 4 are considered rich (line 172-175) and they fully pay health insurance and other social economic needs by themselves. The objective of the study is to determine risk factors for dental caries. 

9. Please describe what analysis was performed in Table 3. What indicator of dental caries was used in comparison with factors associated with dental caries? Is it prevalence? It needs to be described in the manuscript.

Response 9: We are grateful for this comment and we thank the reviewer for point it out. We conducted bivariate analyses through a series of single-predictor binary logistic regression analyses in table 3.

 The indicator of dental caries that we used in comparison with factors associated with dental caries was prevalence. We described it in the manuscript as well (line 205). 

10. The meaning of the sentence in this line of 178 and 179 “Participants who visited dentists after 1 year but less than 5 years” is unclear. 

Response 10: We agree with the reviewer’s comment and we edited the statement as follow: “Participants who visited dentists within 5 years” (line 198)

11. I don't quite understand the analysis method used by the authors in Table 3. If it was a single factor Chi-square analysis, why were HIV-related factors such as viral load didn’t need to be included in this analysis, but only in the regression analysis? If it was the result from a logistic regression, the total associated factors also should be included and the results should be presented in one table including both the Crude Odds Ratio and the Adjusted Odds Ratio.

Response11: We really regret the confusion resulted from table 3 and we thank the reviewer for point it out. Table 3 resulted from bivariate analyses through a series of single-predictor binary logistic regression analyses. Since table 3 compared HIV positive and HIV negative participants, we first considered common factors (all factors not related to HIV infection) for HIV positive and HIV negative persons to be able to compare them. Then a separate analysis (results of table 4) was considered for factors that were unique for HIV negative persons alone (CD4, Viral load, WHO staging for HIV, etc) but we adjusted for all previous common factors in table 3. 

12. The fitting test result of the logistic regression model should be attached at the end of Table 4, so that readers can judge the reliability of the model.

Response 12: We understand and agree with the reviewer comments. We reported Pseudo R squared result as it is a common model fit parameter. 

Fitting test result for table 3 (HIV+): pseudo R2= 0.131 (line 206)

Fitting test result for table 3 (HIV-): pseudo R2= 0.141 (line 207)

Fitting test result for table 4: pseudo R2= 0.174 (line 225)

---

## [Decision Letter · Decision Letter 1]

7 Feb 2023

Prevalence of dental caries and associated risk factors among HIV positive and HIV negative adults at an HIV clinic in Kigali, Rwanda

PONE-D-22-27201R1

Dear Dr. Murererehe,

We’re pleased to inform you that your manuscript has been judged scientifically suitable for publication and will be formally accepted for publication once it meets all outstanding technical requirements.

Kind regards,

Gaetano Isola, Ph.D.

Academic Editor

PLOS ONE

Additional Editor Comments (optional):

The authors have well addressed all comments raised by both reviewers in both round of revision.

Reviewers' comments:

Reviewer's Responses to Questions

**Comments to the Author**

1. If the authors have adequately addressed your comments raised in a previous round of review and you feel that this manuscript is now acceptable for publication, you may indicate that here to bypass the “Comments to the Author” section, enter your conflict of interest statement in the “Confidential to Editor” section, and submit your "Accept" recommendation.

Reviewer #1: All comments have been addressed

2. Is the manuscript technically sound, and do the data support the conclusions?

Reviewer #1: Yes

3. Has the statistical analysis been performed appropriately and rigorously? 

Reviewer #1: I Don't Know

4. Have the authors made all data underlying the findings in their manuscript fully available?

Reviewer #1: Yes

5. Is the manuscript presented in an intelligible fashion and written in standard English?

Reviewer #1: Yes

6. Review Comments to the Author

Reviewer #1: The authors have ably responded to the comments appropriately.

A few more (discretional) issues to consider

1. Terminology. HIV + and HIV – could be changed to HIV infected, or People Living with HIV/AIDS (PLWHA) and HIV could be referred to as HIV uninfected. (lines 157 to 163)

2. In discussion the following options could be considered.

a. HIV infected persons were more likely to live in urban setting – was this low income or higher income urban setting. Could they have moved to low income urban setting to be nearer to health services over the years as opposed to having a higher income. This could also be related to higher education status. (Lines 313 to 317)

b. Women had more prior dental care. It’s not uncommon for women to be more inclined to seek medical attention earlier than men – a trend commonly noted in some populations, rather than women having a tendency to more dental caries. (Lines 306 to 307).

7. PLOS authors have the option to publish the peer review history of their article (what does this mean?). If published, this will include your full peer review and any attached files.

Reviewer #1: **Yes: **Harriet Mayanja-Kizza

---

## [Editor Report · Acceptance letter]

29 Mar 2023

PONE-D-22-27201R1 

Prevalence of dental caries and associated risk factors among People Living with HIV/AIDS  and HIV uninfected  adults at an HIV clinic in Kigali, Rwanda 

Dear Dr. Murererehe:

I'm pleased to inform you that your manuscript has been deemed suitable for publication in PLOS ONE. Congratulations! Your manuscript is now with our production department. 

Kind regards, 

on behalf of

Prof. Gaetano Isola 

Academic Editor

PLOS ONE